# Disentangling Transfer in Continual Reinforcement Learning

**Maciej Wołczyk**[*]
Faculty of Mathematics and Computer Science
Jagiellonian University
Kraków, Poland
maciej.wolczyk@doctoral.uj.edu.pl

**Michał Zając**[*]
Faculty of Mathematics and Computer Science
Jagiellonian University
Kraków, Poland
emzajac@gmail.com

**Razvan Pascanu**
DeepMind
London, UK
razp@google.com

**Łukasz Kuciński**
Polish Academy of Sciences
Warsaw, Poland
lkucinski@impan.pl

**Piotr Miłoś**
Ideas NCBR,
Polish Academy of Sciences,
deepsense.ai
Warsaw, Poland
pmilos@impan.pl

## Abstract

The ability of continual learning systems to transfer knowledge from previously seen tasks in order to maximize performance on new tasks is a significant challenge for the field, limiting the applicability of continual learning solutions to realistic scenarios. Consequently, this study aims to broaden our understanding of transfer and its driving forces in the specific case of continual reinforcement learning. We adopt SAC as the underlying RL algorithm and Continual World as a suite of continuous control tasks. We systematically study how different components of SAC (the actor and the critic, exploration, and data) affect transfer efficacy, and we provide recommendations regarding various modeling options. The best set of choices, dubbed `ClonEx-SAC`, is evaluated on the recent Continual World benchmark. `ClonEx-SAC` achieves $87\%$ final success rate compared to $80\%$ of PackNet, the best method in the benchmark. Moreover, the transfer grows from $0.18$ to $0.54$ according to the metric provided by Continual World.

## 1 Introduction

The ability of continual learning (CL) systems ([17, 22]) to utilize knowledge from previously seen tasks in order to maximize transfer on the current task is a significant challenge for the field. Achieving progress in this area would bring benefits both for real-life applications and multiple machine learning domains [24, 18, 46, 10], including reinforcement learning (RL), as advocated in [47]. In particular, it would constitute a critical step towards making efficient lifelong learning agents a reality.

The goal of this paper is to expand our understanding of transfer and its driving factors in continual reinforcement learning (CRL). As the underlying RL algorithm, we assume soft actor-critic (SAC), see [16], and use Continual World [47] as the suite of continuous control environments. We systematically study the critic and actor networks, the key components of SAC, with regard to their influence on transfer. Similarly, we measure the impact of various choices regarding exploration and buffer data usage. The low-level mechanisms of transfer are not yet fully understood even in the supervised

---

[*]equal contribution

36th Conference on Neural Information Processing Systems (NeurIPS 2022).

learning case [28]. To the best of our knowledge, our work is the first one that undertakes a comprehensive study of this important topic in RL. To this end, we proceed in two stages: exploring a two-task setting and a full continual learning scenario.

We start by investigating a simplified two-tasks setting in Section 4. This allows us to leave out the impact of forgetting, as well as limit the choices regarding exploration and data handling. We use 100 pairs of robotic tasks from the Continual World benchmark. Here, we render two key observations: 1) the role of the critic is the most important for transfer, while exploration and actor play smaller, but non-negligible, parts; 2) contributions of the individual components are mostly independent. Additionally, we show that the concept of feature reuse which is often utilized to explain supervised transfer learning [28, 33] might not be directly applicable in RL.

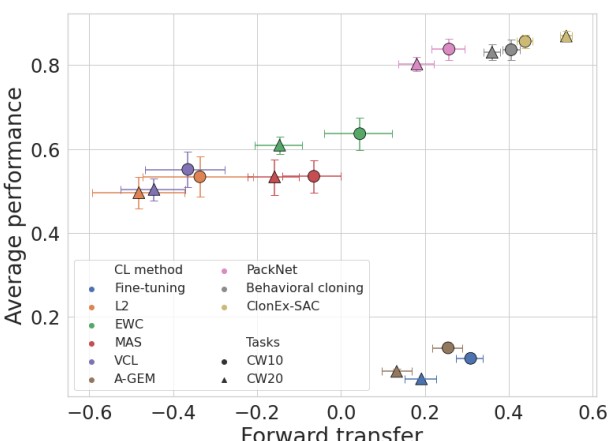

Figure 1: Performance of the `ClonEx-SAC` method compared with competitive baselines, on CW10 and CW20 task sequences. Average performance and forward transfer are shown, together with 90% bootstrap confidence intervals.

In Section 5, we aim to understand new effects which emerge for the full continual learning scenario, typically in longer sequences. In CL context, we need to take into account forgetting, being mindful of the fact that existing CL methods often favor mitigating forgetting at the expense of transfer, see [47]. Main results include 1) reusing policies from previous tasks for exploration considerably improves performance; 2) behavioral cloning to rehearse past tasks is beneficial for both average performance and forward transfer, outperforming other considered methods; 3) regularizing the critic typically does not help for the performance of CL methods.

The result of our comprehensive analysis is a set of general recommendations. We also determine the combination of design choices that outperforms all other options, dubbed `ClonEx-SAC`. This method utilizes behavioral *cloning* to mitigate catastrophic forgetting for the actor. Moreover, at the beginning of each task, `ClonEx-SAC` queries all previous policies, the best of which generates initial *exploration* data. `ClonEx-SAC` achieves 87% final success rate compared to 80% of PackNet, the best method in the Continual World benchmark, see Figure 1. Importantly, we observe a sharp transfer increase from 0.18 to 0.54 in the metric provided in the benchmark. Notably, the value of forward transfer closely matches the reference forward transfer adjusted for exploration, which is a soft upper bound for transfer, as introduced in [47].

## 2 Related work

Continual learning algorithms are often categorized into three classes: regularization-based e.g. [2, 23, 31], parameter isolation e.g. [26] and rehearsal methods e.g. [6, 7]; see also CL survey papers [8, 17, 32]. [44, 4] advocate the need to develop CL methods suitable for reinforcement learning training as a necessary step towards learning artificial intelligent agents to operate in open-ended and changing environments. [22] provides a detailed review of this combination and a taxonomy of possible setups. [47] proposes a sequence of robotic tasks as a benchmark, comparing popular CL methods adapted to RL and advocating for putting more emphasis on transfer.

The authors of [20] show how the synaptic Benna-Fusi model can be added on top of value-based RL methods to mitigate forgetting at both intra- and inter-task scales. A simple approach to cloning policies from previous tasks is employed in [45], and a similar replay strategy has been used in [35]. [21] tackles the case when task boundaries are not provided. Although most of the research is concerned with model-free continual reinforcement learning, an approach to model-based continual RL was presented in [19].

Transfer learning, which focuses on the reuse of machine learning models, has been extremely successful recently. In computer vision, convolutional neural networks [24, 18] and vision transformers [11] pre-trained on large datasets can be repurposed and fine-tuned on the target task. Modern transformer-based models [46, 10] trained on large natural language corpora turned out to be very flexible and can be adapted to diverse downstream tasks with surprising efficiency [34, 25]. General surveys of transfer learning techniques are provided in [52, 41]. Interestingly, recent research [29, 50] suggests that there are still some gaps in our understanding of transfer learning. [29] analyzes the low-level reasons for transfer, exhibiting surprising phenomena such as transfer between datasets with permuted images. [50] performs large-scale experiments investigating representation transfer in a wide variety of visual tasks.

In reinforcement learning scenarios, the structure of the underlying MDP can be exploited to facilitate the transfer. [42, 51, 43, 40] present methods on how to find and use mappings between different domains. [30, 5] apply reward function reshaping. [27, 39] achieve transfer by means of high-level skills and hierarchical RL. Other lines of work exploit the model structure [36, 13] or enforce modularity [3, 9]. In this work, we aim to complement these studies, by focusing on the benefits of reusing neural network parameters, and other choices that exploit the RL structure, like exploration and data rehearsal.

## 3 Background

### 3.1 Continual learning and reinforcement learning

Continual learning tackles the problem of learning in non-stationary settings [8]. Typically, the solution is expected to perform well on all encountered tasks, although various metrics expressing different requirements are formulated. The popular CL desiderata include reducing the *forgetting* on previous tasks and increasing the *forward transfer* on the new tasks, i.e. speeding up the learning by reusing knowledge from previous tasks [12, 47]. Other desiderata focus on limiting resources, such as the number of samples, computation time, model size, or additional memory size. These requirements are often conflicting, so usually some trade-offs have to be made [17, 47, 32].

Combining CL with RL adds another layer of complexity. In this work, we focus on the SAC algorithm [16], which is often considered to be the method of choice for continuous control RL [49, 48, 38]. As an actor-critic algorithm, it is based on the interplay between its two parts, see Section 3.2. This is a fairly complicated algorithmic setup, which presents a number of challenges when used jointly with CL.

In particular, since the optimization of the critic and actor networks is intertwined, it is hard to understand and decouple the impact of individual components. Additionally, because of this interplay, training biases get easily exacerbated, often leading to inferior performance or even a collapse. Another complication is that the training objectives for the actor and the critic are different. The critic minimizes the Bellman error which is known to be a fragile objective [14] susceptible to training biases and might correlate poorly with the value error (which we would like to minimize). As the actor optimizes over predictions of the critic, it might also suffer from these problems, even if less directly. Finally, since the policy and the data we see change during the training, there is an inherent distribution shift present, even within a single task.

### 3.2 SAC

In our study, we focus on soft actor-critic (SAC) [15], an off-policy actor-critic RL algorithm, based on the maximum entropy principle. **The critic** strives to approximate the entropy-corrected $Q$-function under the current policy, optimizing the Bellman error. **The actor** tries to find actions that maximize the $Q$-function. **The replay buffer** holds the seen experience and provides data for the actor and critic updates at each learning step. **The exploration policy** is used to gather data at the beginning of each task for a set number of $K$ steps. By default, in most SAC implementations, this means sampling actions uniformly over the action space.

### 3.3 Continual World

We perform our experiments on the Continual World [47] benchmark. It contains a set of realistic robotic tasks, where a simulated Sawyer robot manipulates everyday objects. The structure of the observation and action spaces remains the same between the tasks; an observation is a 12-dimensional vector describing the coordinates of the robot's gripper and relevant objects. The 4-dimensional action space describes the gripper movement. In training, a dense reward function is used to make the tasks solvable; in evaluation, the binary success metric is used to indicate whether the desired goal has been reached. The tasks are arranged in sequences and training in each task lasts for 1M steps. CW10 sequence contains 10 different tasks arranged in a fixed order. CW20 consists of CW10 repeated twice, allowing to measure how much knowledge can be transferred in case of task repetitions. We use both CW10 and CW20 in our evaluations, as well as shorter sequences containing pairs of tasks from CW10.

### 3.4 Metrics

Following standard practice in continual learning literature, we report average performance and forgetting metrics. We also measure transfer as defined in [47]. Below we briefly recall these three metrics. Assume $p_i(t) \in [0, 1]$ to be the performance (success rate) of task $i$ at time $t$, and that each of the $N$ tasks is trained for $\Delta$ steps, so the total number of steps is $T = N \cdot \Delta$.

**Average performance**    The average performance at time $t$ is defined as $\mathrm{P}(t) := \frac{1}{N} \sum_{i=1}^{N} p_i(t)$. Its final value, $P(T)$, is a scalar summary of the performance and is presented in the result tables.

**Forward transfer**    The forward transfer is computed as a normalized area between the training curve of the measured run and the training curve of a reference curve from training from scratch. Let us denote by $p_i^b \in [0, 1]$ the reference performance. Then the forward transfer on task $i$, $FT_i$, is defined as

$$\mathrm{FT}_i := \frac{\mathrm{AUC}_i - \mathrm{AUC}_i^b}{1 - \mathrm{AUC}_i^b}, \quad \mathrm{AUC}_i := \frac{1}{\Delta} \int_{(i-1) \cdot \Delta}^{i \cdot \Delta} p_i(t)\mathrm{d}t, \quad \mathrm{AUC}_i^b := \frac{1}{\Delta} \int_0^{\Delta} p_i^b(t)\mathrm{d}t.$$

The average forward transfer for all tasks, FT, is defined as $\mathrm{FT} = \frac{1}{N} \sum_{i=1}^{N} \mathrm{FT}_i$.

**Forgetting**    For the task $i$, one can measure a drop in performance after the end of learning on this task as $F_i = p_i(i \cdot \Delta) - p_i(T)$. Forgetting metric is defined as $F = \frac{1}{N} \sum_{i=1}^{N} F_i$.

### 3.5 Experimental setup

We follow the experimental setup from [47]. The actor and the critic are implemented as two separate MLP networks, each with 4 hidden layers of 256 neurons. We refer to the 4 hidden layers as the *backbone* and the last output layer as the *head*. By default, we assume the *multi-head* (MH) setting, where each task has its separate output head, but we also consider the *single-head* (SH) setting, where only a single head is used for all tasks. The SAC exploration phase takes $K = 10k$ steps. All experiments in this paper were performed with 10 different seeds unless noted otherwise. We compute 90% confidence intervals through bootstrapping. More details on the experimental setup can be found in Appendix A.

## 4 Transfer in isolation

In this section, we study *what enables transfer between RL tasks*. We assume a two-task setting, where we measure the forward transfer from the first to the second task, disregarding issues specific to continual learning (e.g. forgetting), which we defer to the next section. We utilize all 100 pairs of CW10 tasks, see Section 3.3, to evaluate the impact of *critic, actor, and exploration* given by SAC.

We will say that the actor or the critic are *carried over* (from the previous tasks) if their parameters are reused as the initialization in the next task; otherwise, the parameters are re-initialized. We also refer to the exploration policy as being carried over, if we use the policy from the previous task (or tasks) to gather the data during the first 10k steps of the SAC exploration phase (see Section 3.2);

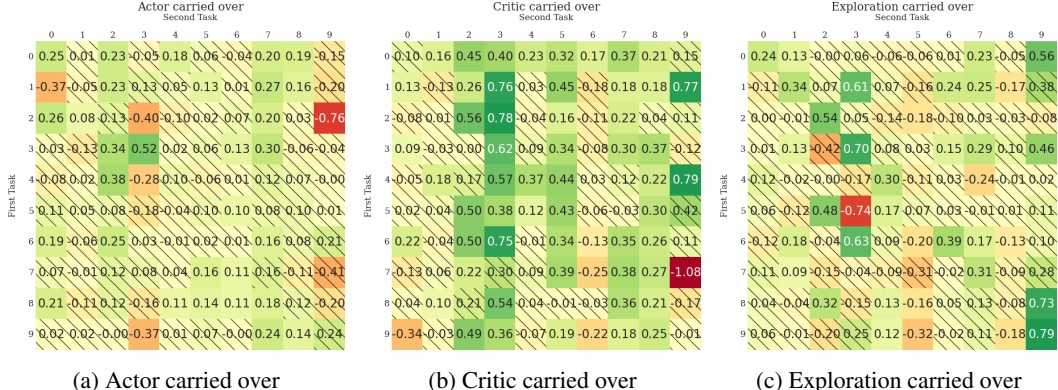

Figure 2: The effect of carrying over different components on the performance on pairs of tasks from CW10. We shade an entry if the $90\%$ confidence interval contains 0, indicating that we cannot be sure whether the component which was carried over makes a difference.

otherwise, a uniform random exploration policy is being used. We use both multi-head (MH) and single-head (SH) settings, with the former being default.

Figure 2 illustrates the impact of the individual components on transfer for each pair. The $(i, j)$-th entry in the matrix contains the forward transfer value when carrying over components from task $i$ to task $j$. Table 1 presents the aggregated statistics from the matrices given in Figure 2: the average FT (including and excluding the diagonal), and the number of pairs with positive, negative, and neutral FT[2]. Table 2 reports the transfer properties for all possible combinations of components present in Table 1, omitting single-head critic (since it performs worse in Table 1).

### 4.1 Carrying over SAC components

From the results presented above, we draw two key observations. First, the role of the critic is the most important for FT, while exploration and actor play smaller, but non-negligible, parts. Second, the components are "transfer independent", in the sense that the transfer of the combination of the components is close to the sum of transfers yielded by each component alone.

The evidence for the first finding is presented in detail for each pair in Figure 2 and summarized in Table 1. More precisely, the average forward transfer across all pairs attributed to carrying over of the critic equals $0.2$ (resp. $0.15$) for MH (resp. SH) setup. This separates the critic from the actor and exploration, which yield (for the default MH setup) $0.06$ and $0.09$, respectively.

The importance of the critic is further emphasized by showing that restraining its learning capabilities, even when the weights are initialized to the parameters learned in the previous task, negatively impacts FT. This is shown in the last row of Table 1, where only the critic's head is allowed to train, while the body of the network is kept frozen and carried over from the previous task. This result goes against our understanding of transfer in supervised learning, where feature reuse is a common technique (e.g. in vision [28, 33]). However, the deterioration in FT can be explained by RL-specific factors. Namely, freezing the backbone can hinder both the policy training (since the mechanics of SAC intertwines actor and critic) and the critic training (due to inflated Bellman errors).

As to the second finding, i.e. the "transfer independence" of the components, the results of the underlying analysis are presented in Table 2. We observe that the reported FT for the combination of components follows closely the sum of FTs for individual components (reported in Table 1). Furthermore, we observe that including all the components results in the highest transfer of $0.35$.

There is a couple of remaining interesting observations. First, Figure 2 exhibits several vertical patterns, meaning that transfer depends more on the second task. Second, the effect on transfer increases on the diagonal, when the exploration is carried over. This seems reasonable since the

---

[2]We say that a pair has positive (resp. negative) FT if the corresponding confidence interval is above (resp. below) 0. Otherwise, we mark it as neutral.

Table 1: Summary of the transfer statistics from the transfer matrices when transferring only a single component. FT and FT (no diag) represent average forward transfer across all pairs with and without considering the diagonal (transfer from a task to the same task), respectively. Subsequent columns denote the number of pairs with the positive, negative, and neutral transfer.

| name | FT | FT (no diag) | # pos. | # neg. | # neutral |
|------|-----|-----|-----|-----|-----|
| Actor (MH) | 0.06 [0.03, 0.10] | 0.05 [0.01, 0.09] | 30 | 5 | 65 |
| Critic (MH) | 0.20 [0.17, 0.23] | 0.19 [0.16, 0.23] | 54 | 5 | 41 |
| Exploration | 0.09 [0.06, 0.13] | 0.06 [0.03, 0.10] | 28 | 9 | 63 |
| Actor (SH) | 0.12 [0.09, 0.15] | 0.12 [0.09, 0.15] | 37 | 1 | 62 |
| Critic (SH) | 0.15 [0.12, 0.18] | 0.13 [0.10, 0.16] | 41 | 19 | 40 |
| Critic (train only head) | -1.29 [-1.33, -1.25] | -1.30 [-1.35, -1.26] | 0 | 100 | 0 |

Table 2: Summary of transfer statistics when multiple components are carried over. We observe that impact of each component is largely independent of other components. That is, FT when carrying over multiple components is close to the sum of FT when carrying over each of them separately.

| name | FT | FT (no diag) | # pos. | # neg. | # neutral |
|------|-----|-----|-----|-----|-----|
| Actor (MH) + Critic (MH) | 0.27 [0.24, 0.30] | 0.25 [0.22, 0.29] | 58 | 4 | 38 |
| Actor (SH) + Critic (MH) | 0.29 [0.26, 0.32] | 0.28 [0.25, 0.31] | 59 | 2 | 39 |
| Actor (MH) + Exp. | 0.16 [0.12, 0.20] | 0.14 [0.10, 0.18] | 39 | 3 | 58 |
| Actor (SH) + Exp. | 0.21 [0.17, 0.24] | 0.18 [0.15, 0.22] | 53 | 0 | 47 |
| Critic (MH) + Exp. | 0.30 [0.27, 0.33] | 0.28 [0.25, 0.31] | 64 | 2 | 34 |
| Actor (SH) + Critic (MH) + Exp. | 0.36 [0.33, 0.38] | 0.33 [0.29, 0.36] | 68 | 0 | 32 |
| Actor (MH) + Critic (MH) + Exp. | 0.35 [0.31, 0.38] | 0.32 [0.29, 0.36] | 70 | 1 | 29 |

policy in the new task is initialized to the already learned policy on the same task. Finally, resetting the head (MH setup) is beneficial in the case of the critic, while it hurts the actor.

## 5 Transfer in continual learning

In Section 4.1, we focused on direct transfer in the two-task setting. Now, we move to the full continual learning scenario, which brings two main differences: 1) we measure the performance of all tasks in the sequence, so forgetting now plays a significant role; 2) typically, we consider longer sequences of tasks of length 10 and 20 (CW10 and CW20, respectively). For longer sequences, forgetting and transfer may have complex mutual interactions. To reduce forgetting, CL methods usually apply some kind of regularization to the model, which in turn may be harmful to transfer. On the other hand, transfer benefits from accumulated knowledge – if forgetting is not mitigated, there might be nothing to transfer from.

We will investigate three main themes. The first one is reusing previous policies for exploration. For long sequences, there are multiple design choices available compared to the two-task scenarios. Secondly, we investigate CL with data reuse, an approach successful in supervised learning. We show that the CRL setup is more complex and requires careful investigation. Finally, given the importance of the critic for transfer (see Section 4), we study whether the critic should be regularized or not, and conclude that typically, the answer is negative.

We study these issues in conjunction with various CL methods: Fine-tuning, Perfect memory, EWC, PackNet, L2, A-GEM, MAS, and VCL. These are standard CL approaches adopted and tested in the RL setting [47], see details in Appendix B. We note that CL methods used here are mostly successful in mitigating forgetting; in this section, we report average performance and forward transfer, deferring forgetting to the Appendices C and E.

### 5.1 Exploration

When using the SAC algorithm, at the beginning of each task, there is a short period of exploration with a random policy, see Section 3.2. The experiments in Section 4.1 showed that the transfer

increases if the policy from the previous task is used instead. Now, we pass from two-task scenarios to longer ones, and analyze the following options for choosing exploration policy, which we call: *random, preceding, uniform-previous*, and *best-return*, and define them as follows. In the first task, we always use a random policy, and assume that the tasks are numbered from 1 to $N$.

Consider now $i \in \{2, \ldots, N\}$. For the *random* strategy, we randomly sample from the action space, which is a default choice for SAC. For the other strategies, at the beginning of each exploration episode, we choose a previous actor head to generate data instead of the random policy. In the case of the *preceding* strategy, we use the $(i-1)$-th actor's head. For *uniform-previous* policy, we use the $j$-th actor's head, where $j := \text{RANDOM\_UNIFORM}(\{1, \ldots, i-1\})$. Finally, in *best-return* strategy, we first try every possible head, and then act using the $j_{\max}$-th actor's head, where $j_{\max} := \text{argmax}_{j \in \{1, \ldots, i-1\}} R_j^i$; $R_j^i$ is the return of the $j$-th head policy on the $i$-th task.

We evaluate how these strategies interact with various CL methods. We pick Fine-tuning, Behavioral cloning, L2, EWC, and PackNet. The results for two well-performing methods, EWC and PackNet, are presented in Table 3, with the rest being deferred to Appendix E. For EWC, choosing any non-random policy significantly improves upon the baseline random strategy. This is particularly visible in the CW20 sequence, which contains repeated tasks, and arguably can benefit more from an informed strategy like best-return. Interestingly, the results for the rather simple uniform-previous approach are quite competitive. We observe increased performance also for other methods except for PackNet, for which effects are negligible.

Table 3: Average performance and forward transfer for different exploration strategies on CW10 and CW20 sequences. Strategies are added on top of EWC and PackNet methods.

| Method, exploration | CW10 perf. | CW10 f. transfer | CW20 perf. | CW20 f. transfer |
|---|---|---|---|---|
| **EWC, random** | 0.63 [0.60, 0.66] | 0.03 [-0.04, 0.09] | 0.60 [0.59, 0.62] | -0.14 [-0.19, -0.09] |
| **EWC, preceding** | **0.70** [0.67, 0.73] | 0.09 [0.03, 0.15] | 0.61 [0.59, 0.64] | -0.14 [-0.19, -0.09] |
| **EWC, uniform-previous** | **0.72** [0.69, 0.75] | **0.24** [0.19, 0.28] | **0.70** [0.68, 0.73] | 0.21 [0.17, 0.25] |
| **EWC, best-return** | **0.70** [0.68, 0.73] | **0.25** [0.21, 0.28] | **0.71** [0.69, 0.73] | **0.28** [0.25, 0.31] |
| **PackNet, random** | 0.84 [0.81, 0.86] | 0.26 [0.22, 0.29] | 0.80 [0.79, 0.82] | 0.18 [0.14, 0.22] |
| **PackNet, preceding** | 0.84 [0.82, 0.85] | 0.24 [0.20, 0.27] | 0.81 [0.80, 0.83] | 0.20 [0.16, 0.24] |
| **PackNet, uniform-previous** | 0.84 [0.81, 0.86] | 0.21 [0.15, 0.26] | 0.80 [0.78, 0.82] | 0.23 [0.18, 0.27] |
| **PackNet, best-return** | 0.85 [0.83, 0.86] | 0.23 [0.20, 0.26] | 0.82 [0.81, 0.83] | 0.23 [0.21, 0.25] |

## 5.2 Data rehearsal

Rehearsal techniques work very well in supervised continual learning [7]. In RL, two major approaches to utilizing previous data have been studied: applying them as offline data using SAC loss, and behavioral cloning of the previous policies. The former, dubbed Perfect memory, was reported to perform poorly [47]. Behavioral cloning achieves more promising results [45, 35]. We study these two approaches with an emphasis on the effects on transfer.

In Perfect memory, all the experiences are kept in the buffer. SAC training is applied to data from the current task and offline data from the previous ones. In Behavioral cloning, an additional small buffer is filled at the end of training on each task. We annotate a subset of samples from the main SAC buffer using the trained actor and critic networks. When training the new task, we sample data from expert buffers and apply auxiliary losses (with different weights), minimizing the KL divergence between current and saved outputs for the actor and L2 distance for the critic; see details in Appendix B.

Firstly, we study the effect of rehearsal on transfer in the two-task scenario, using 100 task pairs from CW10, as in Section 4.1. We observe that using Perfect memory or cloning both actor and the critic has a detrimental effect on transfer, providing more evidence that critic regularization can be catastrophic. On the other hand, cloning only the actor has a neutral effect; we report results for these and more setups in Appendix E. As such, in the remaining Behavioral cloning experiments, we regularize only the actor, unless noted otherwise.

Secondly, we perform experiments on longer sequences, CW10 and CW20; see Table 4. For reference, we include two methods tested in [47], Fine-tuning and PackNet. Fine-tuning achieves the highest transfer and PackNet the highest overall performance out of the methods tested in [47]. Behavioral

Table 4: Average performance and forward transfer for Perfect memory and Behavioral cloning methods, as described in Section 5.2. Fine-tuning and PackNet are included for reference.

| method | CW10 perf. | CW10 f. transfer | CW20 perf. | CW20 f. transfer |
|---|---|---|---|---|
| **Perfect memory** | $0.27$ [0.24, 0.30] | $-1.13$ [-1.23, -1.04] | $0.09$ [0.06, 0.12] | $-1.32$ [-1.41, -1.24] |
| **Behavioral cloning** | **0.84** [0.81, 0.86] | **0.41** [0.38, 0.43] | **0.83** [0.81, 0.85] | **0.36** [0.34, 0.38] |
| **Fine-tuning** | $0.10$ [0.10, 0.10] | $0.31$ [0.27, 0.34] | $0.05$ [0.05, 0.05] | $0.19$ [0.15, 0.23] |
| **PackNet** | **0.84** [0.81, 0.86] | $0.26$ [0.22, 0.29] | $0.80$ [0.79, 0.82] | $0.18$ [0.14, 0.22] |

cloning performs very well. In terms of the average performance, it is on par with PackNet on CW10 and better on CW20. Importantly, it significantly outperforms the baselines in terms of transfer. We can see that Perfect memory works poorly, in line with the existing literature. In Appendix C, we present the results for five other CL methods benchmarked in [47].

In the end, we observe an interesting phenomenon. While behavioral cloning does not improve transfer in two-task scenario, it has a positive effect for the longer sequences. This result hints that perhaps the learner accumulates knowledge of the previous tasks and, thus, can reuse the most relevant parts of the past to improve the training of the current task. Additionally, perhaps behavioral cloning loss acts as a regularizer and helps shape more general features, thus further improving transfer.

### 5.3 Regularizing the critic

This section is devoted to the study of critic regularization in CRL methods. Since in our formulation of the problem, the primary objective of CRL is the final performance of the actor, we have some flexibility in how we treat the critic. We can even completely ignore forgetting in the critic, as recommended in [47]. Other works suggest that regularization might be beneficial [21].

To understand this issue better, we carefully measure the performance while varying the strength of the regularization, by changing the critic regularization coefficients for EWC, L2, and Behavioral cloning. We first find a good value for the actor regularization coefficient, with the critic regularization coefficient being set to $0$. Then, with this value, we perform the sweep over the critic coefficients, covering a wide range from $1 \times 10^{-10}$ to $100$, and run training on the CW10 sequence. For all three methods, we observe that for the smallest values of critic regularization, the performance is similar to the version without critic regularization, and then after some threshold, performance visibly deteriorates. In the case of Behavioral cloning, it drops from $0.82$ (no critic regularization) to $0.77$ (critic regularization coefficient $= 0.001$) and then further, see Table 15. The complete results are presented in Appendix E.3.

This confirms the practical recommendation from [47] to regularize only the actor. One possible explanation is that $TD$-learning used for the critic is highly sensitive to biases introduced by regularization.

## 6 Combining the improvements: `ClonEx-SAC`

Based on the experimental findings presented so far, we propose to combine the discovered enhancements in a simple method for continual reinforcement learning. This method significantly improves the performance in the Continual World benchmark [47]. In particular, we observe a sharp transfer increase to a value that matches a soft upper bound for transfer introduced in [47].

We incorporate the following choices in the proposed method:

- We use behavioral cloning for the actor, which, as we showed in Section 5.2, effectively mitigates forgetting and increases transfer.

- We use best-return exploration, as described in Section 5.1, which efficiently reuses old policy heads for faster exploration.

- As indicated in Section 5.3, we do not use any CL regularization for the critic.

- We use multiple output heads for both actor and critic to profit from transferred representations without introducing too much bias in the new tasks, as discussed in Section 4.1.

We dub the method `ClonEx-SAC` to reflect the usage of the behavioral **clon**ing, improved **ex**ploration, and SAC algorithm.

We compare `ClonEx-SAC` with the behavioral cloning and 7 methods considered in [47], on the CW10 and CW20 sequences. We present results in Figure 1 (see Introduction) and Appendix C. `ClonEx-SAC` achieves $87\%$ final performance compared to $80\%$ of PackNet, the best method in [47].

The forward transfer of `ClonEx-SAC`, improves sharply from $0.19$, the best previous result, to $0.54$. Notably, `ClonEx-SAC`'s result closely matches the reference forward transfer, see below. We conjecture that this excellent transfer is an important factor in the final performance. We also notice that improvements brought separately by behavioral cloning and the best-return exploration strategy work well together.

**Reference forward transfer** (RT) was introduced in [47] as a soft upper bound for transfer. For a sequence of tasks $t_1, \ldots, t_N$, it is defined defined as $\mathrm{RT} := \frac{1}{N} \sum_{i=2}^{N} \max_{j<i} \mathrm{FT}(t_j, t_i)$, where $F(t_j, t_i)$ denotes the forward transfer for the pair of tasks $t_j, t_i$.

Intuitively, $RT$ estimates the level of forward transfer, which could be achieved when a method is able to remember and transfer all meaningful aspects of previously seen tasks. Note that in principle, higher values of $RT$ could still be achievable if the knowledge from the previous tasks is composed. In our setup, the values of $RT$ are $0.44$ for CW10 and $0.55$ for CW20. In both cases, they are closely matched by the forward transfer of `ClonEx-SAC`. We note that our $RT$ values are higher than the one reported in [47], since their work does not take into account the effects of improved exploration.

# 7    Limitations

We are fully aware that our analyses do not cover the entire spectrum of problems that one might be interested in when studying transfer in CRL. Here, we summarize a few limitations of our work:

- We build on top of the SAC algorithm. There is a risk that some of the conclusions from this paper would differ for another choice of the underlying RL method.

- We focus on the Continual World suite. There is a possibility that some of the results from this paper would differ in environments from other domains or with different, potentially structured state spaces.

- `ClonEx-SAC` requires retaining data from previous tasks, which may not always be feasible (e.g., due to privacy concerns).

# 8    Conclusions

In this work, we identify and study some of the key factors contributing to transfer in continual reinforcement learning. In the first part of the study, we focus on the transfer alone, disregarding other CL desiderata, and analyze how different components of the SAC algorithm (actor, critic, exploration) contribute to it. We identify the critic as the leading component.

In the second part, we study further effects that are relevant to the full continual learning setup with long task sequences. In particular, we show that behavioral cloning and reusing previous policies for exploration significantly improve both transfer and the final performance. This leads to a new method, `ClonEx-SAC`, which outperforms considered baselines.

We believe that this work constitutes the first step toward understanding the mechanisms behind transfer in continual reinforcement learning. There are still important issues to be resolved, e.g., pinpointing the exact role of feature reuse or the interplay between transfer and forgetting. We hope that these will be addressed by the community in the future.

## Acknowledgments and Disclosure of Funding

The work of Maciej Wołczyk was supported by the National Centre of Science (Poland) Grant No. 2021/43/B/ST6/01456. The work of Piotr Miłoś was supported by the Polish National Science Center grant UMO-2017/26/E/ST6/00622 and UMO-2019/35/O/ST6/03464. This research was supported by the PL-Grid Infrastructure. Our experiments were managed using `https://neptune.ai`. We would like to thank the Neptune team for providing us access to the team version and technical support.

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
