# OpenReview forum: "Disentangling Transfer in Continual Reinforcement Learning"
_NeurIPS.cc/2022/Conference — NeurIPS 2022 Accept_

### Official Review · Reviewer_uTPf · 2022-07-11

**Rating:** 7
**Confidence:** 3
**Soundness:** 4 excellent
**Presentation:** 4 excellent
**Contribution:** 3 good

**Summary:**

**Summary:** The paper aims to study different factors that improve transfer in continual learning. It focuses on continual world benchmark and SAC as the underlying algorithm. It concludes the following:

  1. Transfer of critic alone matters more than that of actor and exploration. Furthermore, the benefit (i.e. increase in performance and forward transfer) from transferring multiple components is usually close to the sum of benefits gained when those components are transferred alone.

  2. Regularizing actor loss with BC (using expert data from prior tasks) helps and it’s best to not regularize critic loss

  3. Using previously trained policy (as opposed to random policy) for exploration on new tasks helps

The paper uses these findings to come up with a modified form of SAC named ClonEX-SAC which gives state-of-the-art performance on CW10 and CW20 benchmarks.


**Questions:**

N/A

**Limitations:**

The authors were very upfront about the limitations of the work (Section 7) which I appreciated.

**Strengths And Weaknesses:**

**Strength:**
  1. The experimental analysis in the paper is quite thorough. I like the fact that it uses pair of tasks to come up with initial hypothesis and then tests those hypothesis on CW10/CW20 benchmarks.

  2. The final proposed method nicely follows from the analysis done in previous section and is overall simple while still achieving state of the art performance in CW10/CW20

**Weaknesses:**
  1. The paper mostly focuses on SAC as the underlying algorithm and CW10/CW20 benchmarks. Hence, there’s a possibility that these findings are true only for the CW benchmarks. I do appreciate the authors being very upfront about the limitations of the work.

---

> ### Author Response · Authors · 2022-08-02
> **Response to the Reviewer uTPf**
>
> We thank the Reviewer for encouraging and constructive feedback. We are glad that the Reviewer appreciates thorough experimental section, simplicity and usefulness of the final method.
>
> The question concerning other RL algorithms has been on our radar. In this paper, we focused on SAC as a popular and successful RL algorithm and leave the study of other methods for future research. We would like to point out that the scope of experiments here was extensive (over 100K experiments, 12M CPU hours) and exploring the same questions for different RL algorithms was computationally out of reach for us within the scope of this project. Having said that, we can try to speculate that the findings would be similar for the algorithms based on the Q-learning (SAC/DDPG/TD3), while it is less clear about the policy gradient methods (A3C/PPO etc.), as the critic plays a less prominent role there.

---

### Official Review · Reviewer_3Rch · 2022-07-19

**Rating:** 5
**Confidence:** 3
**Soundness:** 3 good
**Presentation:** 3 good
**Contribution:** 2 fair

**Summary:**

The paper presents an empirical evaluation on how to improve knowledge transfer in the continual RL (CRL) setting. The authors consider SAC as the RL algorithm of interest and perform their evaluations on the Continual World benchmark. First, the paper investigates 3 different ways of transferring knowledge obtained by the SAC agent in a 2-task setting. Second, the authors conduct experiments in CRL setting in which they combine SAC with different CL algorithms. They explore different choices for an exploration strategy, data rehearsal and regularising the critic. Finally, they combine all their findings to create a SAC agent which outperforms the competitors.


**Questions:**

q1: Could you briefly list the contributions of this paper?

q2: what knowledge do you think can be transferred between actors? You refer to the expected transfer of feature representation in supervised learning, however, in your setting, the input already provides an abstract representation of the environment.


**Limitations:**

The discussion on the limitations of this approach is adequate.

**Strengths And Weaknesses:**

The paper takes a step in an interesting direction, as knowledge transfer is an important and relatively overlooked property in CL. For the most part, the paper is well written an easy to follow.

In terms of originality, in my opinion, the paper does not introduce novel ideas, but evaluates different combinations of ideas previously found in the literature. The experimental setting and the measurements are also not novel. It would have been helpful if the paper’s claimed contributions were listed in the introduction. Nevertheless, the experimental findings of this paper are interesting and might be beneficial to people doing research in CRL with SAC. I’m not sure how the findings apply to other RL algorithms, aside from SAC, and I did not see a discussion on this.

Overall, while the findings are interesting, I can’t decide if they are significant enough to warrant a recommendation for acceptance.

---

> ### Author Response · Authors · 2022-08-02
> **Response to the Reviewer 3Rch**
>
> We thank the Reviewer for constructive feedback. We are glad that the Reviewer thinks the findings of the paper are interesting and the text is well-written. Below, we try to address the Reviewer’s concerns about the work.
>
> To the best of our knowledge, our paper is the first to shed light on the transfer problem in CRL settings. The transfer mechanisms are not fully understood, even in supervised learning [2,3], and the community has recently advocated for more emphasis on their study [1,4]. The significance of our paper stems from an empirical grounding in CRL with a comprehensive suite of experiments, which we feel will have high utility for the community. Additionally, SAC is a very popular algorithm, and the understanding we gain here should be universally useful. The following are our key contributions:
> 1. Measurement of the relative importance of RL algorithm components (with the critic being the most important) and showing that their contributions are orthogonal. Additionally, we observe that (surprisingly) it is best not to regularize the critic.
> 2. Evaluation of various exploration schemes and showing that using policies from old tasks improves performance on a new task.
> 3. A new method that outperforms all considered baselines in terms of average performance and forward transfer on the CW20 benchmark.
>
> More importantly, our work proves a solid basis for further research, and we hope it will catalyze the community's effort toward understanding transfer mechanisms.
>
> The question concerning transfer between actors is a good one. Even in computer vision, where the idea that reusing “low-level” features (e.g. edge detection) allows for transfer is prevalent, the transfer is a quite complex problem [2]. In our case, there is no clear or intuitive way in which the “low-level” features can be reused; however, the existence of such transfer has already been observed, e.g. in [4]. We speculate that at least two “low-level” factors play a role here: 1) statistics of the neural network weights, i.e., means, variances and higher order moments; 2) features at the early layers of the neural network, constituting reusable components, thus speeding up learning in the consequent tasks. Approaching this question systematically and quantitatively is an interesting future work direction.
>
> Incentivized by the Reviewer’s comments, we performed additional experiments which indicate that reusing features from a different task in a policy can be useful. We used a simplified setup, where we train a linear policy for the Hammer-v1 via Augmented Random Search [5]. This algorithm was chosen as we wanted to verify features usefulness detached from a particular algorithm, and also ARS is a simple, policy-only algorithm. We considered the following variants for the input of the policy:
>
> A) raw features from the environment; input dimension = 12 (performance 0.25),
>
> B) penultimate layer features from the SAC actor trained on the same task, Hammer-v1; input dimension = 256 (performance close to 1.0),
>
> C) penultimate layer features from the SAC actor trained on another task, Shelf-place-v1; input dimension = 256 (performance 0.78),
>
> D) random MLP features (from an untrained network) with the same architecture as in B and C; input dimension = 256 (performance 0.55).
>
> We can see that using features from another task (C) is a clear improvement over both A and D. This could be interpreted as some evidence that the actor network obtains useful features for future tasks. Performance in C is worse than in B because the features useful for another task are not fully aligned with features useful for the Hammer-v1 task; also, as mentioned by the Reviewer Qp95, this can be connected to the data from previous task not being diverse enough.
>
> The question concerning other RL algorithms has been on our radar, and we are upfront about it in the Limitations section. In this paper, we focused on SAC as a popular and successful RL algorithm and leave the study of other methods for future research. We would like to point out that the scope of experiments here was extensive (over 100K experiments, 12M CPU hours) and exploring the same questions for different RL algorithms was computationally out of reach for us within the scope of this project. Having said that, we can try to speculate that the findings would be similar for the algorithms based on the Q-learning (SAC/DDPG/TD3), while it is less clear about the policy gradient methods (A3C/PPO etc.), as the critic plays a less prominent role there.

---

> > ### Author Response · Authors · 2022-08-02
> > **References**
> >
> > [1] Raia Hadsell, Dushyant Rao, Andrei A. Rusu, and Razvan Pascanu. Embracing change: Continual learning in deep neural networks. Trends in Cognitive Sciences, 24(12):1028 – 1040, 2020.
> >
> > [2] Behnam Neyshabur, Hanie Sedghi, and Chiyuan Zhang. What is being transferred in transfer  learning? Advances in neural information processing systems, 33:512–523, 2020.
> >
> > [3] Aniruddh Raghu, Maithra Raghu, Samy Bengio, and Oriol Vinyals. Rapid learning or feature  reuse? towards understanding the effectiveness of maml. arXiv preprint arXiv:1909.09157,  2019.
> >
> > [4] Maciej Wołczyk, Michał Zając, Razvan Pascanu, Lukasz Kucinski, and Piotr Miłos. Continual world: A robotic benchmark for continual reinforcement learning. Advances in Neural Information Processing Systems, 34, 2021.
> >
> > [5] Horia Mania, Aurelia Guy, Benjamin Recht. Simple random search provides a competitive approach to reinforcement learning. arXiv preprint arXiv:1803.07055, 2018.

---

> ### Author Response · Authors · 2022-08-08
> **Follow up**
>
> Dear Reviewer 3Rch,
>
> Thank you again for your review. We wanted to follow up and see if we have addressed your concerns. Please let us know if you have further questions.

---

### Official Review · Reviewer_Qp95 · 2022-08-02

**Rating:** 6
**Confidence:** 3
**Soundness:** 2 fair
**Presentation:** 2 fair
**Contribution:** 2 fair

**Summary:**

​
The authors investigate specific design choices of SAC in affecting transfer efficacy in continual learning on the Continual World benchmark. They then take their findings to achieve state of the art results on Continual World using a relatively simple set of extensions upon SAC.

**Questions:**

How do the authors think these results will transfer to different types of observation spaces? E.g. image-based RL?

**Limitations:**

Sufficient limitations coverage.

**Strengths And Weaknesses:**

## Strengths
​
**Experiment Thoroughness and Analysis:** The authors perform a great job of summarizing their detailed findings intuitively and concisely in the text, and presenting trends across experiments instead of attempting to detail every experiment.
​
**Writing/Clarity:** The paper is written well, and contains a logical flow that makes it easy and pleasing to read.
​
**Motivation and findings:** The paper contributes useful knowledge about continual learning in RL, and is an interesting read and its results should be useful for the community.
​
## Weaknesses
​
**Environment-dependent results:** As the authors allude to in the limitations section of the paper, they demonstrate their results on only the Continual World suite of environments. CW seems to serve as good testbed for the experiments they performed, but the authors should test on at least another continual learning task (e.g Procgen) to show that their findings can also hold in another environment suite (without needing to re-perform most of their testing experiments).
​
**Detailing specific failure cases in the main paper:** The paper results would be more clear if the authors detailed/analyzed some individual specific failure cases in the main paper. For example, why was task 7→9 transfer with the critic so poor in Fig2b?
​
**Minor Issues or comments:**
​
- L81: [28] analyses → [28] analyzes
- In L183-190, when explaining why the deterioration in forward transfer occurs when reusing the features only, it could be also because there is not enough diverse data when just transferring from one task to another in the experiment the authors perform. This is  in contrast to feature transfer in vision, for example, where pre-training on something like Imagenet allows the network backbone to learn very general image-features.
- It would be nice to have a figure of the CW benchmark, to help contextualize the results, in the main paper.

---

> ### Author Response · Authors · 2022-08-02
> **Response to the Reviewer Qp95**
>
> We thank the Reviewer for the constructive feedback. We feel encouraged that the Reviewer values the findings, usefulness, and writing of the paper. Below we answer the questions raised by the Reviewer.
>
> *Evaluation limited to Continual World*
>
> We would like to point out that the scope of the performed experiments has already been extensive (over 100K experiments, 12M CPU hours), and exploring the same questions for different environments has been computationally out of reach for us within the scope of this project. Continual World suite is relevant for real-world robotics scenarios, therefore we felt providing results on this benchmark is a good trade-off between usefulness for the community and computational feasibility.
>
> *Different types of observation spaces (e.g. image-based)*
>
> This is an interesting question, however in our work we wanted to focus on the problem of transfer in RL for low-dimensional observation spaces that are common in robotics tasks. In image-based environments, transfer of skills or features useful for object manipulation (which is not well understood) would be entangled with transfer of computer vision features (which has already been studied, see e.g. [1]). By focusing primarily on simple observation spaces we can highlight the first of these problems. In the end, we find both questions very interesting, but since our experiments have already been quite extensive and we’ve got limited computational resources, we haven’t pursued different observation spaces directly. However, we speculate that most of the key findings would still hold, e.g. exploration from previous tasks being helpful, or behavioral cloning loss working well, as these do not seem to be closely tied to the input form. As for carrying over the critic having the most impact, we think it would still be true, but we anticipate that the transfer values for carrying over network parameters (both actor and critic) would be higher. This is because, in the case of image observations, the additional knowledge about building compressed representations from raw pixel data is present. We think a careful study of this question could be an interesting direction for future work.
>
> *The figure of the CW benchmark*
>
> We include Figure 5 in the supplementary material, containing information about the Continual World benchmark. This figure will be included in the main text for the camera ready version. We also fix the typo in the main text mentioned by the Reviewer.
>
>
> [1] Behnam Neyshabur, Hanie Sedghi, and Chiyuan Zhang. What is being transferred in transfer  learning? Advances in neural information processing systems, 33:512–523, 2020.

---

> ### Author Response · Authors · 2022-08-08
> **Follow up**
>
> Dear Reviewer Qp95,
>
> Thank you again for your review. We wanted to follow up and see if we have addressed your concerns. Please let us know if you have further questions.

---

### Meta-Review · Area_Chair_fgjB · 2022-09-09

**Recommendation:** Accept
**Confidence:** Certain

**Metareview:**

All reviewers appreciated the importance of investigating continual reinforcement learning, and the throughout experiments conducted in the paper. While the paper does not introduce new methods, it evaluates existing methods and offers surprising insights. While the experiments focus on a single algorithm and environment suite, these insights are still valuable to the community. For these reasons, I recommend acceptance.

**Award:**

No

---

### Decision · Program_Chairs · 2022-09-14

Accept